# Pregnancy Outcomes and SARS-CoV-2 Infection: The Spanish Obstetric Emergency Group Study

**DOI:** 10.3390/v13050853

**Published:** 2021-05-07

**Authors:** Sara Cruz Melguizo, María Luisa de la Cruz Conty, Paola Carmona Payán, Alejandra Abascal-Saiz, Pilar Pintando Recarte, Laura González Rodríguez, Celia Cuenca Marín, Alicia Martínez Varea, Ana Belén Oreja Cuesta, Pilar Prats Rodríguez, Irene Fernández Buhigas, María Victoria Rodríguez Gallego, Ana María Fernández Alonso, Rocío López Pérez, José Román Broullón Molanes, María Begoña Encinas Pardilla, Mercedes Ramírez Gómez, María Joaquina Gimeno Gimeno, Antonio Sánchez Muñoz, Oscar Martínez-Pérez

**Affiliations:** 1Department of Gynecology and Obstetrics, Puerta de Hierro University Hospital of Majadahonda, 28222 Majadahonda, Spain; saracruz.gine@yahoo.es (S.C.M.); beenpar@yahoo.es (M.B.E.P.); 2Fundación de Investigación Biomédica, Puerta de Hierro University Hospital of Majadahonda, 28222 Majadahonda, Spain; 3Department of Gynecology and Obstetrics, University Hospital 12 de Octubre, 28041 Madrid, Spain; paolacp1993@gmail.com; 4Department of Gynecology and Obstetrics, La Paz University Hospital, 28046 Madrid, Spain; alejandra_as@hotmail.com; 5Department of Gynecology and Obstetrics, Gregorio Marañon University Hospital, 28007 Madrid, Spain; ppintadorec@yahoo.es; 6Department of Gynecology and Obstetrics, Hospital Alvaro Cunqueiro, 36213 Vigo, Spain; laura_gr_@hotmail.com; 7Department of Gynecology and Obstetrics, Regional Hospital of Málaga, 29010 Málaga, Spain; celia.cuenca.sspa@juntadeandalucia.es; 8Department of Gynecology and Obstetrics, La Fe University and Polytechnic Hospital, 46026 Valencia, Spain; martinez.alicia.v@gmail.com; 9Department of Gynecology and Obstetrics, Hospital del Tajo, 28300 Aranjuez, Spain; anaoreja@yahoo.es; 10Department of Gynecology and Obstetrics, QuirónSalud Dexeus University Hospital, 08028 Barcelona, Spain; pilpra@dexeus.com; 11Department of Gynecology and Obstetrics, Torrejón University Hospital, 28850 Torrejón de Ardoz, Spain; ibuhigas80@gmail.com; 12Department of Gynecology and Obstetrics, San Millán-San Pedro Hospital Complex, 26006 Logroño, Spain; marivirg80@gmail.com; 13Department of Gynecology and Obstetrics, Torrecárdenas University Hospital, 04009 Almería, Spain; anafernandez.alonso@gmail.com; 14Department of Gynecology and Obstetrics, Santa Lucía University Hospital, 30202 Cartagena, Spain; rocio.lopez.perez@gmail.com; 15Department of Gynecology and Obstetrics, Puerta del Mar University Hospital, 11009 Cádiz, Spain; jrbroullon@gmail.com; 16Maternal-fetal Medicine Unit, Department of Gynecology and Obstetrics, La Mancha Centro General Hospital, 13600 Alcázar de San Juan, Spain; mercebon@hotmail.com; 17Maternal-fetal Medicine Unit, Department of Gynecology and Obstetrics, Reina Sofia University Hospital, 14004 Córdoba, Spain; qgimenog@gmail.com; 18Department of Gynecology and Obstetrics, Ciudad Real University Hospital, 13005 Ciudad Real, Spain; asanchezm@sescam.jccm.es

**Keywords:** SARS-CoV-2, coronavirus, COVID-19, pregnancy, delivery, perinatal outcomes, premature birth, maternal complications

## Abstract

Pregnant women who are infected with SARS-CoV-2 are at an increased risk of adverse perinatal outcomes. With this study, we aimed to better understand the relationship between maternal infection and perinatal outcomes, especially preterm births, and the underlying medical and interventionist factors. This was a prospective observational study carried out in 78 centers (Spanish Obstetric Emergency Group) with a cohort of 1347 SARS-CoV-2 PCR-positive pregnant women registered consecutively between 26 February and 5 November 2020, and a concurrent sample of PCR-negative mothers. The patients’ information was collected from their medical records, and the association of SARS-CoV-2 and perinatal outcomes was evaluated by univariable and multivariate analyses. The data from 1347 SARS-CoV-2-positive pregnancies were compared with those from 1607 SARS-CoV-2-negative pregnancies. Differences were observed between both groups in premature rupture of membranes (15.5% vs. 11.1%, *p* < 0.001); venous thrombotic events (1.5% vs. 0.2%, *p* < 0.001); and severe pre-eclampsia incidence (40.6 vs. 15.6%, *p* = 0.001), which could have been overestimated in the infected cohort due to the shared analytical signs between this hypertensive disorder and COVID-19. In addition, more preterm deliveries were observed in infected patients (11.1% vs. 5.8%, *p* < 0.001) mainly due to an increase in iatrogenic preterm births. The prematurity in SARS-CoV-2-affected pregnancies results from a predisposition to end the pregnancy because of maternal disease (pneumonia and pre-eclampsia, with or without COVID-19 symptoms).

## 1. Introduction

With more than 126,000,000 confirmed cases, the SARS-COV-2 pandemic is a life-threatening health problem, especially in high-risk individuals [1].

Due to the physiological changes of pregnancy, pregnant women are more vulnerable to respiratory infections [2] and for this reason, pregnancy should be considered a high-risk condition during the COVID-19 pandemic. 

We currently know that pregnant women are at an increased risk of developing more severe COVID-19 symptoms compared to the general population, but also may suffer increased adverse perinatal outcomes [3]. Compared to non-infected pregnant women, SARS-CoV-2-positive pregnant women have increased odds of maternal death, of needing admission to the intensive care unit (ICU), and of preterm birth, leading to more neonatal intensive care unit admissions [4,5]. How obstetric intervention may influence the clinical course of the disease in these patients has also been described [6]. 

The Spanish Obstetric Emergency Group (SOEG), which has one of the largest series of SARS-CoV-2-infected pregnant women in the world, has contributed to the previous findings. With the present study, which includes a complete cohort of infected patients and a concurrent sample of non-infected patients and encompasses the first two high-incidence waves of SARS-CoV-2 (1 March to 5 May 2020, and 14 July to 5 November 2020) [7], we aim to better understand the relationship between maternal infection and perinatal outcomes, with a focus on preterm birth and the underlying medical and interventionist factors.

## 2. Materials and Methods

This was a multicenter prospective study of a cohort of SARS-CoV-2-infected pregnant women registered consecutively by the SOEG in 78 hospitals (Appendix A) [8]. All procedures were approved by the Drug Research and Clinical Research Ethics Committee of Puerta de Hierro University Hospital (Madrid, Spain) on 23 March 2020 (protocol registration number, 55/20). Each collaborating center subsequently obtained protocol approval locally (ethics committees of the participant hospitals listed in the Appendix A). The registry protocol is available on ClinicalTrials.gov, identifier: NCT04558996. Upon recruitment, mothers consented to participate in the study by either signing a document when possible, or by giving permission verbally, which was recorded in the patient’s chart in the electronic clinical recording system. Ethics committees approved the possibility of verbal consent during the first three months of the pandemic given the contagiousness of the disease and the lack of personal protection equipment. Afterwards, written consent (using the patient consent form) was collected from every patient who had previously given permission verbally.

A specific database was designed for recording information regarding SARS-CoV-2 infection in pregnancy, and the lead researcher for each center entered the data after delivery. We developed an analysis plan using recommended contemporaneous methods and followed existing STROBE guidelines for cohort studies (Appendix A) [9].

During the period of the study, from 26 February to 5 November 2020, we selected all SARS-CoV-2-positive obstetric patients detected by testing suspicious cases that came into hospital due to compatible COVID-19 symptoms and by universal screening for a SARS-CoV-2 infection at admission to the delivery ward (starting on 1 April 2020). A SARS-CoV-2 infection was diagnosed by a positive double-sampling polymerase chain reaction (PCR) from nasopharyngeal swabs. The patients of the cohort were classified as asymptomatic and symptomatic, with the latter stratified into three groups: mild–moderate symptoms (cough, anosmia, fatigue/discomfort, fever, dyspnea, etc.), pneumonia, and complicated pneumonia/shock (with ICU admission and/or mechanical ventilation and/or septic shock).

Non-infected patients were those defined as having a negative PCR at admission to delivery, and with no symptoms pre- or postpartum. In order to have a representative non-infected comparison group, each center provided between one and two PCR-negative asymptomatic pregnancies per infected mother by providing either a standardized randomization table or by selecting negative pregnancies that delivered immediately before or after each infected mother. This method was deployed to adjust for center conditions and management at the time of delivery, and to decrease the risk of selection bias.

Information regarding the demographic characteristics of each pregnant woman, comorbidities, and previous and current obstetric history was extracted from the clinical and verbal history of the patient. Subsequently, age and race were categorized following the classifications used by the CDC (Centers for Disease Control and Prevention) [10]. For perinatal events, we recorded gestational age at delivery, the onset of labor and the type of delivery, preterm delivery (below 37 weeks), premature rupture of membranes (PROM), preterm premature rupture of membranes (PPROM), ICU admission, obstetrical complications (pre-eclampsia, hemorrhagic and thrombotic events), stillbirth, and maternal mortality. Neonatal data included a five-minute Apgar score, umbilical artery pH, birth weight, neonatal intensive care unit (NICU) admission, and neonatal mortality. Definitions of clinical and obstetric conditions followed international criteria [11,12,13]. Preterm deliveries were classified as spontaneous (including those resulting from a PPROM), induced labor/C-section due to PPROM, and iatrogenic (due to maternal or fetal reasons). Patients were followed until six weeks postpartum. Neonatal events were recorded until 14 days postpartum.

The numerical variables of maternal age, gestational age at delivery, gestational age at PPROM, days in ICU, and birth weight of newborns were tested for normal distribution using the Kolmogorov–Smirnov test. Descriptive data of the infected cohort and the non-infected comparison group are presented as median (interquartile range, IQR) for the numerical variables (mentioned above), or number (percentage) for the categorical variables (the remaining ones). *p*-values of the univariable analysis (comparison between infected and non-infected) were obtained by Mann–Whitney’s U test for the numerical variables and by the Pearson’s chi-squared test or the Fisher’s exact test for the categorical variables. Statistical tests were two-sided and were performed with SPSS V.20 (IBM Inc., Chicago, IL, USA); a *p*-value below 0.05 was considered statistically significant.

In order to elucidate the reasons underlying iatrogenic delivery (no PPROM) among SARS-CoV-2-infected singleton preterm deliveries, the influence of COVID-19 mild–moderate symptoms, pneumonia (including complicated pneumonia), pre-eclampsia (moderate and severe) and their interactions were analyzed with multivariable logistic regression modeling, deriving the adjusted odds ratio (aOR) with a 95% confidence interval (95% CI) of these factors. These variables were selected after verifying their statistical association with iatrogenic delivery among the SARS-CoV-2-infected singleton preterms. Modeling was performed after excluding pregnancies with missing data. The regression analysis was carried out using the lme4 package in R, version 3.4 (RCoreTeam, 2017) [14]. The multivariable logistic regression model created was as follows:(1)Iatrogenic delivery(a)=COVID symptoms(b)+pre−eclampsia(c)+interaction of both

(a) 2 categories: non-iatrogenic delivery (reference category) and iatrogenic delivery among SARS-CoV-2-infected singleton preterms; (b) 3 categories: asymptomatic (reference category), mild–moderate symptoms, and pneumonia; (c) 2 categories: absence of pre-eclampsia (reference category) and presence of moderate/severe pre-eclampsia.

## 3. Results

### 3.1. Main Results

#### 3.1.1. General Data

During the study period, 2954 patients were recorded in the 78 participating hospitals and analyzed: 1347 pregnant women in the infected cohort and 1607 in the non-infected comparison group (Figure 1).Of the 1347 positive pregnancies, 51.1% (*n* = 688) were asymptomatic at delivery while 48.9% (*n* = 659) showed symptoms.Among symptomatic patients, 70.9% (467/659) showed mild–moderate symptoms, 25.2% (166/659) pneumonia and 3.9% (26/659) complicated pneumonia/shock (with ICU admission and/or mechanical ventilation and/or septic shock).

#### 3.1.2. Baseline and Pregnancy Characteristics

The infected cohort showed a significantly higher proportion of Latin American and Black ethnicities (*p* < 0.001) compared to the non-infected group (Table 1).Maternal age distribution differed between the infected cohort and the non-infected group (*p* < 0.001), being more skewed to the extremes among infected patients (higher proportion of patients under 24 and above 35 years old).

#### 3.1.3. Maternal and Neonatal Outcomes

In the SARS-CoV-2-infected cohort, gestational age at delivery was significantly lower (*p* < 0.001) and the onset of labor was less spontaneous (*p* < 0.001) compared to non-infected pregnancies (Table 2). In addition, C-section rate was higher in infected patients (27.7% vs. 20.4% non-infected, *p* < 0.001).A higher rate of premature rupture of membranes was observed in the SARS-CoV-2 cohort, both when we analyzed globally (PROM: 15.5% vs. 11.1%, *p* < 0.001) and in those less than 37 weeks (PPROM: 2.8% vs. 1.4%, *p* = 0.012).More preterm deliveries (<37 weeks of gestational age) were observed in the SARS-CoV-2-infected cohort (11.1% vs. 5.8%; OR 2.00, 95% CI 1.53–2.62; *p* < 0.001) mainly due to an increase in iatrogenic preterm births, that is, due to medical reasons different from PPROM, as nearly half of preterm births among positive pregnancies were iatrogenic (47.7% vs. 21.3% of preterm births among non-infected; OR 3.37, 95% CI 1.87–6.05; *p* < 0.001).Infected women were more frequently admitted to the ICU before and/or after delivery (2.7% vs. 0.1% non-infected, *p* < 0.001).Women infected with SARS-CoV-2 who developed pre-eclampsia met the criteria for severe pre-eclampsia significantly more than those who were not infected (40.6% vs. 15.6%; OR 3.69, 95% CI 1.62–8.39; *p* < 0.001), while in the latter, the percentage of moderate pre-eclampsia is higher.Higher rates of venous thrombotic events (pulmonary embolism (*p* = 0.003) and disseminated intravascular coagulation (*p* = 0.043)) were observed among infected pregnant women.No differences were noted between the infected cohort and the non-infected group regarding hemorrhagic events.There were two deaths recorded in the SARS-CoV-2-infected cohort versus none in the non-infected group.Higher rates of stillbirths as well as of NICU admissions were observed in the SARS-CoV-2-infected cohort; lower birth weight of newborns from infected mothers was also observed (Table 2).

#### 3.1.4. Reasons for Iatrogenic Delivery among SARS-CoV-2-Infected Singleton Preterm Deliveries 

Among the SARS-CoV-2-infected pregnancies, there was a total of 149 preterm deliveries of which 138 were singletons. The multivariable logistic regression modeling results showed that the following conditions significantly increased the risk of interventionism in preterm deliveries among these patients: pneumonia (aOR 10.83, 95% CI 3.82–34.15; *p* < 0.001), pre-eclampsia (aOR 9.38, 95% CI 1.69–74.76; *p* = 0.016), and pre-eclampsia with COVID-19 mild–moderate symptoms (aOR 15.00, 95% CI 1.90–316.47; *p* = 0.022). 

## 4. Discussion

In this multicenter prospective study, we investigated the association between SARS-CoV-2 infections and obstetric and neonatal outcomes. We found out that pregnant women with a SARS-CoV-2 infection had more premature rupture of membranes, more preterm births and, therefore, their neonates had more NICU admissions, compared to the pregnant women who were not infected [5,15]. The higher risk of premature rupture of membranes (overall as well as preterm) observed in the infected cohort can be explained by the fact that infections in pregnancy may be associated with this condition by various mechanisms, such as the activation of inflammation [16].

When the reasons for preterm births were analyzed in depth, it was observed that the proportion of preterm births resulting from PPROM (both spontaneous and induced/C-section due to this outcome) did not significantly differ between infected (37/149, 24.8%) and non-infected (23/94, 24.5%) mothers (*p* = 0.949). However, it was the medical intervention due to maternal disease that explained the decision to prematurely end the pregnancy; obstetrical interventionism in order to improve the mothers’ health conditions was the main factor for the increased rate of preterm deliveries among the SARS-CoV-2-positive women. It was observed that, not the fact of being infected, but the development of pneumonia or pre-eclampsia (with or without COVID-19 symptoms) was the cause of the increased iatrogenic prematurity in SARS-CoV-2-infected pregnancies. 

Our findings are in line with those previously reported by a study carried out in asymptomatic pregnant women, where an increased risk of PROM was observed among SARS-CoV-2-infected patients when compared to non-infected patients, while this was not the case for preterm delivery [5]. This difference in preterm delivery risk between their study and our study, as explained above, is because preterm delivery is associated with maternal disease manifested in symptomatic patients. This confirms the hypothesis that many obstetric outcomes are related to maternal COVID-19 symptomatology.

The risk of pre-eclampsia was similar for infected and non-infected patients; however, those infected mothers who developed these disorders ended up with severe pre-eclampsia, rather than moderate cases as in the non-infected group. In this association between a SARS-CoV-2 infection and severe pre-eclampsia, a synergistic effect of both factors should not be ruled out [17,18]. However, it must be noted that a severe pre-eclampsia diagnosis is based on hypertensive and biochemical alterations (such as increased lactate-dehydrogenase, thrombocytopenia, and elevated liver enzymes) that can be mixed up with the ones observed in COVID-19 in the general population, apart from the inflammatory status present in both conditions (COVID-19 and pre-eclampsia). Therefore, we must bear in mind that there could be an overestimation of cases of severe pre-eclampsia in the infected cohort since the analytical signs of COVID-19 could have been interpreted as alterations due to pre-eclampsia instead.

No differences were noted between the infected cohort and the non-infected comparison group regarding obstetric hemorrhagic events, while a higher incidence of venous thrombotic events was noted in our SARS-CoV-2-infected pregnancies (1.5%, compared to 0.2% in non-infected), which can be explained by the hemostatic and thromboembolic complications reported in COVID-19 [19]. Even so, the extended heparin prophylaxis policy, which was established in April 2020, may have decreased the expected venous thromboembolism and pulmonary embolism rates in infected patients [20,21]. On the other hand, disseminated intravascular coagulation cases corresponded to the SARS-CoV-2-infected cohort, and this was the underlying cause of a maternal death.

As a limitation of this study, it should be highlighted that symptomatic patients are over-represented in our study population since not all participating hospitals had a universal antenatal screening program for SARS-CoV-2 infections (so only identified symptomatic cases by passive surveillance), or implemented the program later. 

Moreover, the data point to an increased risk of iatrogenic preterm delivery in SARS-CoV-2-infected mothers who developed pneumonia together with pre-eclampsia, but the small number of patients who met these criteria may have penalized the power of analysis. Another limitation of our study is the absence of an in-depth analysis of the biochemical results of the patients who developed pre-eclampsia.

Among the strengths of our study is the large cohort of SARS-CoV-2-positive deliveries (1347 from 78 centers across Spain). In addition, the SARS-CoV-2-negative comparison group was selected from the same centers where the infected mothers delivered and within the same timeframe in order to have similar conditions, thereby minimizing selection and performance biases. We acknowledge as a limitation the absence of the complete screened cohort. However, the concurrent method applied for the selection of a non-infected group (subsample of the screen-negative cohort from all 78 hospitals that had PCR-positive mothers) allowed for a comparison unaffected by the differences in time of exposure and outcome assessment. Therefore, we believe our findings are trustworthy, and the multicenter nature of the study adds to its generalizability.

## 5. Conclusions

Pregnant SARS-CoV-2-infected patients are a population at risk of suffering preterm births, mainly due to iatrogenic deliveries in women with pneumonia and/or pre-eclampsia. Venous thromboembolism and disseminated intravascular coagulation were more frequent in SARS-CoV-2-infected pregnancies.

There is an urgent need for an in-depth analysis of the influence of SARS-CoV-2 infection on the development of pre-eclampsia, and of the risk factors for ICU admittance of pregnant women infected with SARS-CoV-2.

## Figures and Tables

**Figure 1 viruses-13-00853-f001:**
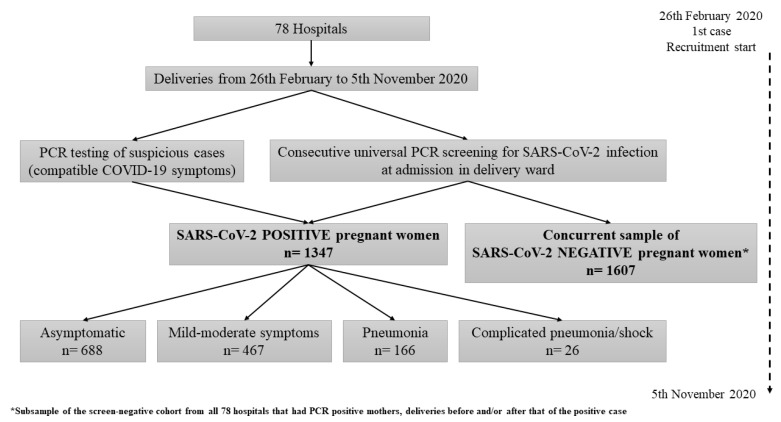
Flow chart of the study data.

**Table 1 viruses-13-00853-t001:** Demographic characteristics, comorbidities, and current obstetric history of the study participants (*n* = 2954).

Number		Infected Cohort	Non-Infected Group	*p*-Value
		1347	1607	
**Maternal Characteristics**				
Maternal age (years; median/IQR)		33 (28–37)	33 (29–36)	0.739
Age Range	18–24	183/1336 (13.7)	165/1585 (10.4)	0.001 *
	25–34	633/1336 (47.4)	850/1585 (53.6)	
	35–49	520/1336 (38.9)	570/1585 (36.0)	
Ethnicity	White European	785/1344 (58.4)	1243/1599 (77.7)	<0.001 *
	Latino American	374/1344 (27.8)	155/1599 (9.7)	
	Black non-Hispanic	35/1344 (2.6)	21/1599 (1.3)	
	Asian non-Hispanic	40/1344 (3.0)	41/1599 (2.6)	
	Arab	110/1344 (8.2)	139/1599 (8.7)	
Nulliparous		516/1333 (38.7)	644/1596 (40.4)	0.366
Smoking ^a^		131/1290 (10.2)	193/1505 (12.8)	0.028 *
**Maternal Comorbidities**				
Obesity (BMI > 30 kg/m^2^)		245/1306 (18.8)	249/1515 (16.4)	0.105
Cardiovascular comorbidities	Baseline heart disease ^b^	15/1316 (1.1)	11/1528 (0.7)	0.241
	Pre-pregnancy HBP	19/1304 (1.5)	17/1514 (1.1)	0.431
Pulmonary comorbidities	Chronic pulmonary disease (not asthma)	3/1316 (0.2)	2/1532 (0.1)	0.667
	Asthma	52/1312 (4.0)	52/1528 (3.4)	0.428
Hematologic comorbidities	Chronic hematologic disease	21/1312 (1.6)	10/1526 (0.7)	0.016 *
	Thrombophilia	25/1310 (1.9)	22/1532 (1.4)	0.325
	Antiphospholipid syndrome	7/1308 (0.5)	8/1524 (0.5)	0.970
Chronic kidney disease		5/1313 (0.4)	5/1528 (0.3)	1.000
Chronic liver disease		11/1319 (0.8)	8/1536 (0.5)	0.305
Rheumatic disease		11/1314 (0.8)	16/1524 (1.0%)	0.560
Diabetes mellitus		26 (1.9)	28 (1.7)	0.704
Depressive syndrome		15/1302 (1.2)	17/1516 (1.1)	0.939
**Current Obstetric History**				
Multiple pregnancies		25 (1.9)	34 (2.1)	0.615
Threatened abortion		41/1275 (3.2)	43/1,545 (2.8)	0.501
High-risk chromosomal abnormality screening		31/1288 (2.4)	37/1544 (2.4)	0.986
High-risk pre-eclampsia screening		69/1149 (6.0)	68/1438 (4.7)	0.150
Positive ultrasound prematurity screening		16/1132 (1.4)	30/1411 (2.1)	0.180
Gestational diabetes		97/1309 (7.4)	136/1584 (8.6)	0.247
Intrauterine growth restriction		48/1290 (3.7)	44/1566 (2.8)	0.170
Pregnancy-induced hypertension ^c^		50 (3.7)	55 (3.4)	0.672

Data are shown as *n* (% of total with data), except where otherwise indicated. BMI: body mass index; HBP: high blood pressure; * statistically significant differences; ^a^ current smoker and ex-smoker; ^b^ including congenital heart disease, not hypertension; ^c^ hypertension + pre-eclampsia.

**Table 2 viruses-13-00853-t002:** Maternal and neonatal outcomes of the study participants (*n* = 2954).

Number		Infected Cohort	Non-Infected Group	*p*-Value
		1347	1607	
**PERINATAL OUTCOMES**				
Gestational age at delivery (weeks + days; median/IQR)		39 + 3 (38 + 2–40 + 3)	39 + 5 (38 + 6–40 + 4)	<0.001 *
Onset of labor	Programmed C-section	142 (10.5)	85 (5.3)	<0.001 *
	Spontaneous	699 (51.9)	1000 (62.2)	
	Induced	506 (37.6)	522 (32.5)	
Type of delivery	Cesarean	373 (27.7)	328 (20.4)	<0.001 *
	Vaginal	832 (61.8)	1044 (65.0)	
	Operative vaginal	142 (10.5)	235 (14.6)	
PROM		209 (15.5)	179 (11.1)	<0.001 *
PPROM		37 (2.8)	23 (1.4)	0.012 *
Gestational age at PPROM (weeks + days; median/IQR)		35 + 0 (33 + 6–35 + 6)	35 + 1 (34 + 6–36 + 3)	0.308
Gestational age range at delivery	<28 weeks	10 (0.7)	7 (0.4)	<0.001 *
	28 to <32 weeks	21 (1.6)	8 (0.5)	
	32 to <37 weeks	118 (8.8)	79 (4.9)	
	≥37 weeks	1198 (88.9)	1513 (94.2)	
Preterm deliveries (<37 weeks of gestational age)		149 (11.1)	94 (5.8)	<0.001 *
Spontaneous delivery (including PPROM)		58/149 (38.9)	62/94 (66.0)	
Induced /C-section due to PPROM		20/149 (13.4)	12/94 (12.8)	<0.001 *
Iatrogenic delivery (no PPROM)		71/149 (47.7)	20/94 (21.3)	
Causes of preterm iatrogenic delivery:				
COVID-19 mild–moderate symptoms		15/71 (21.1)	0/20 (0.0)	
Pneumonia ^a^ (alone)		27/71 (38.0)	0/20 (0.0)	
Pre-eclampsia ^b^ (alone)		5 ^c^/71 (7.0)	6/20 (30.0)	
COVID-19 mild-moderate symptoms + pre-eclampsia ^b^		7/71 (9.9)	0/20 (0.0)	
Pneumonia ^a^ + pre-eclampsia ^b^		7/71 (9.9)	0/20 (0.0)	
Other		10/71 (14.1)	14/20 (70.0)	
Admitted in ICU ^d^		36 (2.7)	2 (0.1)	<0.001 *
Days in ICU (median/IQR)		12 (8.5–17)	3 (3–3)	0.128
Hemorrhagic events		70 (5.2)	89 (5.5)	0.682
Abruptio placentae		12 (0.9)	7 (0.4)	0.123
Postpartum hemorrhage		61 (4.5)	86 (5.4)	0.306
Pre-eclampsia		69 (5.1)	64 (4.0)	0.137
Severe pre-eclampsia		28/69 (40.6)	10/64 (15.6)	0.001 *
Admitted in ICU ^a^		10/28	0/10	
Invasive ventilation		4/28	0/10	
Moderate pre-eclampsia		41/69 (59.4)	54/64 (84.4)	0.001 *
Thrombotic events		7 (0.5)	2 (0.1)	0.089
Deep venous thrombosis		10 (0.7)	1 (0.1)	0.003 *
Pulmonary embolism		4 (0.3)	0 (0.0)	0.043 *
Disseminated intravascular coagulation				
Stillbirth		10 (0.7)	3 (0.2)	0.023 *
**MATERNAL MORTALITY**		2 (0.1)	0 (0.0)	0.208
**NEONATAL DATA**				
Apgar 5 score <7		20/1335 (1.5)	21/1597 (1.3)	0.674
Umbilical artery pH < 7.10		40/1081 (3.7)	46/1248 (3.7)	0.985
Birth weight (grams; median/IQR)		3240 (2890–3550)	3290 (2970–3600)	0.001
Admitted in NICU		137 (10.2)	39 (2.4)	<0.001 *
Neonatal mortality		6 (0.4)	2 (0.1)	0.153

Data are shown as *n* (% of total with data), except where otherwise indicated; * statistically significant differences; PROM: premature rupture of membranes; PPROM: preterm premature rupture of membranes; ICU: intensive care unit; NICU: neonatal intensive care unit; ^a^ both pneumonia and complicated pneumonia/shock; ^b^ both moderate and severe pre-eclampsia; ^c^ asymptomatic patients; ^d^ before and/or after delivery.

## Data Availability

The data presented in this study are available on request from the corresponding author. The data are not publicly available due to the multicenter nature of the study.

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
