# Peer review of "Pregnancy Outcomes and SARS-CoV-2 Infection: The Spanish Obstetric Emergency Group Study"

_viruses, 2021, doi:10.3390/v13050853_

Round 1
Reviewer 1 Report
The manuscript focuses on the Spanish obstetric and neonatal outcomes concerning COVID-19 and pregnancy. These results seem to be very useful in the age of SARS-CoV2 pandemic.
However, there are some limitations:
The authors informed (lines 224-225): "When the reasons for preterm births were deeply analyzed, it was observed that the proportion of spontaneous preterm births or those resulting from PPROM did not significantly differ between infected and non-infected mothers.", whereas I am able to find "Spontaneous delivery (including PPROM)" 58/149 (38.9) vs. 62/94 (66.0) in the table 2. Please provide "p-values" for the discussed causes of preterm deliveries.
Could you check "p-value <0.001" regarding "gestational age at delivery" - "39+3 (38+2–40+3)" for the infected group vs. "39+5 (38+6–40+4)" for the control group.
Please provide data regarding incidences of preterm deliveries, i.e. sub-categories based on gestational age including "extremely early preterm birth". Please compare them between the studied groups
Please add the birth weight of newborns in the infected and non-infected groups.
Please consider replacing the word "eutocic" (Table 2) with another one, for example "natural".
There are some abbreviations in the manuscript, which should be expanded, i.e. CDC (line 116), PROM, PPROM, ICU, NICU (in the Table 2).
Please use "Instruction for Authors" for all references.
Author Response
Response to Reviewer 1 Comments
The manuscript focuses on the Spanish obstetric and neonatal outcomes concerning COVID-19 and pregnancy. These results seem to be very useful in the age of SARS-CoV2 pandemic.
Response 1: Thank you.
However, there are some limitations:
The authors informed (lines 224-225): "When the reasons for preterm births were deeply analyzed, it was observed that the proportion of spontaneous preterm births or those resulting from PPROM did not significantly differ between infected and non-infected mothers.", whereas I am able to find "Spontaneous delivery (including PPROM)" 58/149 (38.9) vs. 62/94 (66.0) in the table 2. Please provide "p-values" for the discussed causes of preterm deliveries.
Response 2: We agree with the reviewer that this sentence may be confusing. Here, we referred to preterm deliveries due to PPROM (both spontaneous and induced/C-section due to this outcome), which is 37/149 (24.8%) among infected patients and 23/94 (24.5%) among non-infected (p=0.949). We have reworded the sentence in order to make it clearer (lines 266-268, highlighted copy).
Could you check "p-value <0.001" regarding "gestational age at delivery" - "39+3 (38+2–40+3)" for the infected group vs. "39+5 (38+6–40+4)" for the control group.
Response 3: We double checked the gestational age at delivery (median and IQR) for both groups and the p-value (Mann-Whitney U test) and these are correct (p<0.001).
Please provide data regarding incidences of preterm deliveries, i.e. sub-categories based on gestational age including "extremely early preterm birth". Please compare them between the studied groups.
Please add the birth weight of newborns in the infected and non-infected groups.
Response 4: Thank you for the suggestion; we have added the following sub-categories in order to better describe the distribution of preterm deliveries: <28 weeks (extremely early preterm birth), 28 to <32 wees, 32 to <37 weeks and ≥37 weeks (Table 2). We have also added the birth weight of newborns in the infected and non-infected groups (Table 2 and lines 234-235, highlighted copy).
Please consider replacing the word "eutocic" (Table 2) with another one, for example "natural".
Response 5: We have replaced the word “eutocic” with “vaginal” (Table 2).
There are some abbreviations in the manuscript, which should be expanded, i.e. CDC (line 116), PROM, PPROM, ICU, NICU (in the Table 2).
Response 6: Abbreviations expanded as suggested (line 138 and foot note at Table 2, highlighted copy).
Please use "Instruction for Authors" for all references.
Response 7: We have corrected all references according to "Instruction for Authors".

Reviewer 2 Report
This is an observational, multicenter study, in which 2,954 patients were recorded in the 78 participating hospitals. It consists of a large cohort of 1,347 SARS-CoV-2 positive deliveries, of which 659 were symptomatic and a good number of non-infected women as a comparison group. Despite the methodological limitations present in a "real life cohort", the study provides important and pertinent information on pregnancy outcomes and SARS-CoV-2 infection. The multicenter nature of the study allows its generalizability.
Methods
- Since the authors do not clearly state the design of the study, there is no uniformity in terms throughout the text. I suggest to standardize the terms from the objective, in the introduction, to the analysis plan and the text of the results and discussion, when referring to the study design (It seems to me that it is a cohort study or at least, it was analyzed as such).
- (Lines 127-132) The analysis plan is described as a standard, not adapted to the study itself. The authors could cite variables from the study and how the tests were used for specific comparisons. It is not clear how the variables were selected for the multivariate analysis, or how the model was created.
Results
- Please consider presenting the 95% CI in Table 2, if you agree.
- I suggest that the variables that made up the regression model are presented (in the text of the results).
- It would be interesting if the Supplementary Table S1, which presents the List of hospitals members of the Spanish Obstetric Emergency Group included in this study, also shows the number of positive and negative women participants, per hospital.
- On line 207 the authors report that "Among SARS-CoV-2 infected pregnancies, there was a total of 219 singleton preterm deliveries ", but Table 2 shows only 149 preterm deliveries among SARS-CoV-2 infected pregnancies. Please check that these numbers are correct.
Discussion:
- On line 215, there instead of saying ... we analyzed the relationship between SARS- 215 CoV-2 infection and ... I suggest saying ... “we investigated the association”…
- (Lines 242-252) This paragraph appropriately discusses the possibility that abnormal biochemical tests in women with Covid, caused by the disease itself, may have influenced the definition of eclampsia, in order to overestimate this diagnosis. On the other hand, perhaps the authors could also discuss the role of inflammatory status that can be present in both conditions (covid and pre-eclampsia), and about the role of inflammation in both diseases.
- (Line 255) ... while a high incidence of venous thrombotic events was noted in our SARS-CoV-2 infected pregnancies (1.5%, compared to 0.2% in non-infected) ...
switch to: … a higher incidence…
Conclusions
- What recommendations, for research or for assistance, would the authors make, based on their results and conclusions? Consider making one or more recommendations, if you find it pertinent.
Author Response
Response to Reviewer 2 Comments
This is an observational, multicenter study, in which 2,954 patients were recorded in the 78 participating hospitals. It consists of a large cohort of 1,347 SARS-CoV-2 positive deliveries, of which 659 were symptomatic and a good number of non-infected women as a comparison group. Despite the methodological limitations present in a "real life cohort", the study provides important and pertinent information on pregnancy outcomes and SARS-CoV-2 infection. The multicenter nature of the study allows its generalizability.
Response 1: Thank you.
Methods
- Since the authors do not clearly state the design of the study, there is no uniformity in terms throughout the text. I suggest to standardize the terms from the objective, in the introduction, to the analysis plan and the text of the results and discussion, when referring to the study design (It seems to me that it is a cohort study or at least, it was analyzed as such).
Response 2: We agree with the reviewer; this is a cohort study (cohort of SARS-CoV-2 positive pregnant women, as every positive patient identified in the collaborating hospitals was included in the study; exposure=SARS-CoV-2 infection) with a negative comparison group (concurrent sample of SARS-CoV-2 negative pregnant women, extracted from the screen-negative cohort form all hospitals that had PCR positive mothers). We have standardized the terms throughout the manuscript as suggested.
- (Lines 127-132) The analysis plan is described as a standard, not adapted to the study itself. The authors could cite variables from the study and how the tests were used for specific comparisons. It is not clear how the variables were selected for the multivariate analysis, or how the model was created.
Response 3: We have clarified the statistical analysis as suggested (lines 150-176, highlighted copy).
Results
- Please consider presenting the 95% CI in Table 2, if you agree.
Response 4: We have considered it, but after incorporating the suggestions of other reviewers as well, we have decided to add 95% CI because there would be too much information in Table 2 and this could confuse the readers.
- I suggest that the variables that made up the regression model are presented (in the text of the results).
Response 5: This information has been added to the materials and methods section (lines 170-176, highlighted copy) according to the reviewer’s comment above.
- It would be interesting if the Supplementary Table S1, which presents the List of hospitals members of the Spanish Obstetric Emergency Group included in this study, also shows the number of positive and negative women participants, per hospital.
Response 6: We agree with the reviewer that this information would be interesting for the readers; however and at this stage, this information (especially that of positive patients) cannot be disclosed, according to the authorship policy within our team.
- On line 207 the authors report that "Among SARS-CoV-2 infected pregnancies, there was a total of 219 singleton preterm deliveries ", but Table 2 shows only 149 preterm deliveries among SARS-CoV-2 infected pregnancies. Please check that these numbers are correct.
Response 7: Thank you very much for the comment; it was actually a typo. There were a total of 149 preterm deliveries (138 singleton and 11 twins) among SARS-CoV-2 infected pregnancies. We have corrected this mistake (line 248, highlighted copy).
Discussion:
- On line 215, there instead of saying ... we analyzed the relationship between SARS- 215 CoV-2 infection and ... I suggest saying ... “we investigated the association”…
Response 8: Corrected as suggested (lines 256-257, highlighted copy).
- (Lines 242-252) This paragraph appropriately discusses the possibility that abnormal biochemical tests in women with Covid, caused by the disease itself, may have influenced the definition of eclampsia, in order to overestimate this diagnosis. On the other hand, perhaps the authors could also discuss the role of inflammatory status that can be present in both conditions (covid and pre-eclampsia), and about the role of inflammation in both diseases.
Response 9: Thank you very much for the suggestion; we have added this point in the discussion (lines 293-294, highlighted copy).
- (Line 255) ... while a high incidence of venous thrombotic events was noted in our SARS-CoV-2 infected pregnancies (1.5%, compared to 0.2% in non-infected) ...
switch to: … a higher incidence…
Response 10: Corrected as suggested (line 299, highlighted copy).
Conclusions
- What recommendations, for research or for assistance, would the authors make, based on their results and conclusions? Consider making one or more recommendations, if you find it pertinent.
Response 11: We consider that there is an urgent need for an in-depth analysis of the influence of SARS-CoV-2 infection on the development of pre-eclampsia, and of the risk factors for ICU admittance of pregnant women infected with SARS-CoV-2. We have included this reflection in the conclusions (lines 333-335, highlighted copy).

Reviewer 3 Report
The manuscript describes the impact on pregnancy outcomes of asymptomatic and symptomatic SARS-CoV-2 infection compared with uninfected controls. The abstract carefully notes that infected women with gestational hypertensive disorders were more likely to have met the criteria for severe pre-eclampsia than uninfected women. In turn having pre-eclampsia was one of the two main contributors to the doubling in the rate of pre-term birth. The discussion concerning the interpretation of blood investigations including thrombocytopenia and abnormal liver function tests in women with both Covid-19 and pre-eclampsia is thus important. At present the abstract points to higher incidence of severe pre-eclampsia whilst the discussion is more cautious and suggests an over diagnosis of severe pre-eclampsia. Are there any additional data that could address this point? Can the abstract be re-worded to better reflect the discussion?
Author Response
Response to Reviewer 3 Comments
The manuscript describes the impact on pregnancy outcomes of asymptomatic and symptomatic SARS-CoV-2 infection compared with uninfected controls. The abstract carefully notes that infected women with gestational hypertensive disorders were more likely to have met the criteria for severe pre-eclampsia than uninfected women. In turn having pre-eclampsia was one of the two main contributors to the doubling in the rate of pre-term birth. The discussion concerning the interpretation of blood investigations including thrombocytopenia and abnormal liver function tests in women with both Covid-19 and pre-eclampsia is thus important. At present the abstract points to higher incidence of severe pre-eclampsia whilst the discussion is more cautious and suggests an over diagnosis of severe pre-eclampsia. Are there any additional data that could address this point? Can the abstract be re-worded to better reflect the discussion?
Response 1: Thank you very much for your comments. Unfortunately, we do not have additional data for addressing this point yet; although this is one of our priority objectives, we are still partially limited by the sample size. On the other hand, we have reworded the abstract in order to be more cautious in this point, and in line with the discussion (lines 62-65, highlighted copy).

Round 2
Reviewer 1 Report
Dear Authors,
Thank you for attempting to address my concerns.
I accept the manuscript in the present form.